# Tribological Properties of Ti-TiC Composite Coatings on Titanium Alloys

**DOI:** 10.3390/ma15248941

**Published:** 2022-12-14

**Authors:** Ivan G. Zhevtun, Pavel S. Gordienko, Dmitriy V. Mashtalyar, Yuriy N. Kulchin, Sofia B. Yarusova, Valeria A. Nepomnyushchaya, Zlata E. Kornakova, Sofia S. Gribanova, Danil V. Gritsuk, Alexander I. Nikitin

**Affiliations:** 1Institute of Chemistry, Far Eastern Branch, Russian Academy of Sciences, 690022 Vladivostok, Russia; 2Institute of Automation and Control Processes, Far Eastern Branch, Russian Academy of Sciences, 690091 Vladivostok, Russia; 3Department of Nuclear Technology, Institute of Science-Intensive Technologies and Advanced Materials, Far Eastern Federal University, 690922 Vladivostok, Russia

**Keywords:** titanium alloys, titanium carbide, composite coatings, tribological behavior

## Abstract

The application of titanium and its alloys under friction conditions is severely restricted, owing to their poor wear resistance. The paper presents the results of studies of the composition, microstructure, and tribological properties of Ti-TiC-based composite coatings formed on titanium alloys by the electroarc treatment in an aqueous electrolyte using a graphite anode. It has been found that TiC grains have a different stoichiometry and do not contain oxygen. The grain size varies from hundreds of nanometers to tens of micrometers, and the micro-hardness of the treated surface reached the value of 29.5 GPa. The wear resistance of the treated surface increased approximately 40-fold, and the friction coefficient with steel decreased to 0.08–0.3 depending on the friction conditions. The formation of a composite material based on Ti-TiC will contribute to the effective protection of titanium alloys from frictional loads in engineering.

## 1. Introduction

Titanium alloys are extensively used in aircraft and spacecraft construction, shipbuilding, and many other fields because of their important properties, such as high specific strength, corrosion resistance, nonmagnetic character, and low density [1,2,3,4,5,6,7].

One of the features of the surface properties of titanium alloys that distinguishes them from alloys based on many metals consists of their high surface reactivity contributing, on the one hand, to low welding load upon friction and high values of friction coefficient and wear rate [8,9,10,11,12,13] and, on the other hand, to high affinity for oxygen and, as a result, to formation of thin oxide films, which prevents the adhesion of conventional lubricants to the surface [8,14,15]. This inherent titanium feature causes exceptionally poor anti-friction properties of its alloys and makes impossible their application in friction units of machines and mechanisms without special pretreatment.

There are at least three different approaches in numerous studies in the field of reducing the friction coefficient and improving the anti-friction properties of titanium and its alloys: search of efficient lubricants [15,16,17,18], change of the composition by alloying various metals [19,20], and the one to which the greatest number of works are devoted—surface treatment of titanium alloys. The diffusion hardening of the surfaces of titanium alloys is carried out by saturating the surface layer of the specimens with oxygen, boron, nitrogen, carbon, silicon, etc. [7,21,22]. Coatings on the metal surface are produced by electroplating and chemical methods and precipitation from molten solutions or vapors [7,23]. Surface mechanical attrition treatment promotes cold-hardening and grinding of the surface layer, thus increasing the micro-hardness and wear resistance of titanium alloys [24]. Methods of surface treatment based on the exposure to plasma, such as electrospark deposition (ESD) [25], plasma electrolytic oxidation (PEO) [26], and laser cladding [27,28], are quite effective to improve anti-friction properties. However, most of these methods are not extensively used in practice because of high energy consumption, complicated multistage sequence of operations, or expensive equipment. Some aspects of obtaining composite materials with high mechanical properties are considered in the works [29,30].

Earlier [31], a simple and effective method of improving the anti-friction properties of titanium alloys consisting of arc discharge surface treatment with a graphite anode in an aqueous electrolyte was used. As a result of such treatment, a surface layer of TiC phase is formed to a depth of up to 2 mm in titanium alloy, thereby improving the anti-friction properties of the alloys and their resistance to oxidation. The electrolyte is used in this treatment as a protective medium preventing oxidation of the treated bulk of titanium alloy and providing its rapid cooling. The present work was devoted to study the composition and microstructure of composite coatings based on Ti-TiC and their effect on the tribological characteristics of titanium alloys to provide the theoretical basis for preparation and engineering applications of Ti-TiC-based coatings.

## 2. Materials and Methods

### 2.1. Materials and Treatment

Parallelepiped-shaped samples of a size of 40 × 16 × 3 mm made of titanium alloy VT1-0 (cathode) were treated through electroarc discharge using graphite electrodes (anode) in 0.1–0.2% NaCl aqueous solution (Figure 1). To form a uniform TiC layer, electrodes were moved relatively to each other at a constant rate. The interelectrode gap was not larger than 1 mm, the direct current was 70–90 A, and the anode diameter was d_a_ = 4 mm. The duration of electroarc impact on the surface in local points was 2–4 s. The current source was a TIG160s welding machine.

Thereafter, the samples were grinded in a Tegramin 25 automatic polishing–grinding machine using silicon carbide abrasive paper (grit sizes of 400, 600, and 800); no more than 0.3 mm of the surface was removed. To reveal the microstructure, different methods were used: selective etching in Kroll’s reagent, hydrochloric acid, and nitric acid and fine polishing using abrasive paper (grit sizes 1000 and 1200) and diamond paste of 7/5.

### 2.2. Composition and Structural Analysis

The surface phase composition was estimated using a Bruker D8 ADVANCE X-ray diffractometer with CuK_α_ radiation; X-ray powder diffraction patterns were identified by the EVA program with a PDF-2 powder database. The microstructure and morphology were investigated using a Hitachi S5500 high-resolution scanning electron microscope with a Thermo Scientific energy-dispersive analyzer, a ZeisEVO 40XVP scanning electron microscope with an INCA 350 energy analyzer. For optical microscopic analysis, a METAM LV-41 light microscope (LOMO, Saint-Petersburg, Russia) was used. Raman spectra in local areas of the surface were obtained on a WiTec alpha 500 confocal Raman microscope: laser wavelength 532 nm, signal acquisition time 0.5 s, averaged over 10 spectra, and the measurement error 4 cm^−1^.

### 2.3. Mechanical and Tribological Evaluation

The surface microhardness was determined by means of a PMT-3M microhardness tester (LOMO, Russia) with a Vickers indenter at a load of 1.5 N. Wear tests (Figure 2a) were carried out on a specialized “pin-on-plate” tester with counterbody (bearing steel ShH15) reciprocating motion relative to the fixed sample made on a special order by Avtovaz (Tolyatti, Russia). These systems applied a vertical load to the contact surfaces, parallelepiped-shaped counterbody of a size of 10 × 10 × 4 mm perpendicular to the parallelepiped-shape sample and sliding direction, so that total contact area of the couple—40 mm^2^—did not increase during the test. Test parameters were as follows: counterbody stroke length 15 mm, sliding speed 750 mm/s, shaft rotation frequency 1470 ± 70 min^−1^, and dry friction (without lubrication). The test duration τ and the normal load P were varied: mode 1—τ = 600 s, P = 30 N; mode 2—τ = 1200 s, P = 50 N. The wear was determined by measuring the mass loss of the samples in an electronic balance with an accuracy of 0.1 mg. The data reflected the average value and standard deviation of three wear tests.

The tribological performance of the treated surface was evaluated using a “ball-on-disk” test on a Tribometer TRB friction tester (Figure 2b). The counterbody was a steel ball (Ac100Cr6) with a diameter of 6 mm. Tests were carried out in two stages:(1)At a constant load of 10 N and variable sliding velocities—0.05, 0.1, and 0.2 m/s;(2)At a constant sliding velocity of 0.1 m/s and variable loads—5, 10, and 20 N. Sliding distance—100 m. The friction was dry in air. The disc-shaped samples were used.

The counterbody materials and test conditions were selected to focus on fundamental friction and wear processes of both titanium and composite coating, so the results may not be related to any specific engineering application.

## 3. Results and Discussion

### 3.1. Composition and Structure

The typical XRD patterns of the samples’ surfaces after electroarc treatment are shown in Figure 3. Prior to grinding, five sharp characteristic peaks at 2θ = 36, 42, 61, 72, and 76°, corresponding to the (111), (200), (220), (311), and (222) planes of titanium carbide phase, respectively, are observed along with the peaks attributed to the titanium and rutile phases [32].

The authors of [33] show that when such conditions of treatment are used, the alloy molten bulk is not oxidized and rutile is formed on the surface during the sample cooling at the time when the arc discharge is stopped: in aqueous electrolyte, the treated alloy bulk is quenched. During the subsequent sample grinding, a thin rutile layer is removed, but Ti and TiC remained. The typical SEM images of treated surface before and after grinding are shown in Figure 4.

The element composition determined by energy-dispersive spectroscopy at various local sections on the treated surface before and after grinding is shown in Table 1. These data corroborate the formation of the titanium carbide phase in the titanium alloy bulk under TiO_2_ film.

It can be concluded that in the process of arc discharge surface treatment with a graphite anode using an aqueous electrolyte, the titanium carbide phase, which does not contain oxygen, is formed in the bulk of the cathodically polarized titanium alloy. Oxygen is present only in the composition of oxides on the surface before grinding. According to Table 1, the average ratio of titanium and carbon Ti/C = 2/1, i.e., the volume fraction of titanium is about two times greater than that of carbon. As was shown in [31,34], this was a consequence of formation of the heterogeneous microstructure consisting of finely dispersed titanium carbide particles randomly distributed in the titanium matrix.

To identify and study the composite microstructure of the treated surface, the selective etching technique in various solutions was applied: after removing one of two phases (Ti or TiC), the carbide grains morphology and size and features of their formation were established in the remaining second phase. The typical fragments of the sample surface upon the use of various solutions and fine polishing are shown in Figure 5. The found grain size of TiC in all cases ranged from hundreds of nanometers to tens of micrometers.

Figure 5a–e show the titanium matrix after etching TiC grains in Kroll’s reagent (Figure 5a,b) or nitric acid (Figure 5c;–e). The chemical composition of the surface, according to the energy-dispersive analysis, was 95–99 at. % titanium; the remaining part was carbon. On the contrary, when hydrochloric acid or fine polishing is used (Figure 5f–i), the matrix was removed, but carbide grains, in which the carbon content varied in a fairly wide range from 18 to 43 at. %, were retained.

As was established in studies of the microstructure, in the course of withdrawal from the ‘central’ area undergoing direct impact of the arc discharge, the TiC grains’ shape changed from the markedly grainy with an average grain size of 1–10 µm (Figure 5c,e,g,i) to the dendrite one (Figure 5a,d,h). TiC dendrites are of a width of 5–10 µm and of a length of up to 150 µm. Dendritic microstructure is observed only in samples treated for a sufficiently long time (at least 3 s) and must be explained by the emergence of a temperature gradient between central and peripheral areas that is preserved for a period sufficient for directed grain growth. It is worth mentioning that similar TiC dendrites were observed in [27] after laser treatment of the titanium alloy surface with injection of carbide particles: according to the authors, these dendrites provided the high abrasive-wear resistance of the surface.

Note that in the analysis of X-ray patterns of treated samples a shift of the peaks corresponding to titanium carbide in different cases is observed, which is associated with changes in the lattice constant. These changes in the lattice parameters of titanium carbide can be induced by the presence of both carbon vacancies and impurities [35]. The lattice constant of the sintered TiC phase in different cases corresponds to both stoichiometric titanium carbide without oxygen and other impurities (a_0_ = 4.326 Å) and non-stoichiometric TiC or that containing impurities (a_0_ = 4.3300 Å, 4.317 Å). However, during the energy-dispersive analysis of the surface, only Ti and C without impurities were identified. Hence, this discrepancy can be explained by variations in the stoichiometry of the sintered TiC phase.

This conclusion is supported by the Raman data on the selected surface areas (Figure 6). The spectral peaks at 251, 420, and 614 cm^−1^ correspond to the TiC phase [36]. As was noted in [37], the stoichiometric TiC had no Raman active vibrational modes and all Raman peaks appeared because of carbon vacancies and corresponded to TiCx, where x < 1; with decreasing x, the Raman peaks broadened. Therefore, the presence of distinct peaks establishes the non-stoichiometric composition of the formed titanium carbide. Furthermore, Figure 6 shows that there is carbon on the sample surface (peaks at 1339 and 1583 cm^−1^ [36]). The fact that free carbon is not revealed using XRD gives reason to believe that it is present as an amorphous phase. The presence of free carbon can be explained by the conditions of arc processing, under which carbon transferred from the anode does not completely react with titanium and, therefore, some amount of carbon will inevitably remain on the surface in a free form.

### 3.2. Micro-Hardness

The micro-hardness of the surface after grinding could fall into a broad range—from 7.6 to 29.5 GPa with numerous intermediate values. Such values scattering is the result of both randomly located TiC grains in the composite layer and different stoichiometry of grains themselves: the micro-hardness of pure titanium carbide (30 GPa) is known [35] to decrease as a function of bonded carbon content.

### 3.3. Tribological Performance

Average values of change in the sample weights Δm during the tests on the “pin-on-plate” tester are shown in Figure 7. Since absolute values of the surface wear changed differently depending on the test load and duration, we considered the values of initial samples wear relatively to that of the treated samples Δm_init_/Δm_tr_. This relative increase in the wear resistance remains approximately constant under different test conditions and is about 40. It is evident that the wear resistance of the samples is increased through formation of a composite structure based on Ti-TiC.

During the samples tests on the tribological “ball-on-disk” tester after sliding distance of 100 m, the friction tracks formation on the titanium sample surface is observed. The process of friction is accompanied by a transfer of titanium on a harder steel counterbody (Figure 8a,b), which is typical for the process of titanium friction [8,9,14,16]. Similar tests on the treated samples show that the friction track on the surface is scarcely visible, but counterbody wear is observed (Figure 8c,d).

Friction coefficient changes during the process of initial samples friction in a pair with a steel ball at different sliding velocities and vertical loads are shown in Figure 9.

At sliding velocities of 0.1 and 0.2 m/s (at the same load of 10 N), friction pairs demonstrate similar friction behavior (Figure 9a): first, the friction coefficient increases and attains a maximum (around μ = 0.5); thereafter, some decrease in μ down to values of 0.45 is observed. The increase in the friction coefficient at the test start occurs, probably, because of the increase in both pair contact area (formation of friction track) and local temperature initiating oxidation processes and the phase transition in titanium: α (hexagonal close-packed) → β (body-centered cubic). Such a phase transition is known [38,39] to contribute to an abrupt increase in the friction coefficient. Note that there is a phenomenon of solid-phase welding and material transfer on a mating surface when the titanium friction occurs with any harder material, so that titanium rubs on titanium and the friction coefficient of such a pair approximates to the friction coefficient of the titanium-titanium pair, which is in the range 0.45–0.55 depending on the friction conditions [1]. Further gradual decrease in the friction coefficient is possible because of the formation of titanium oxides preventing the welding. However, a similar decrease was observed in vacuum [39] and was explained by the decrease in the strain-hardening role. The authors of [40] relate the latter fact to recrystallization processes in the surface layer of titanium samples.

In contrast, at lower sliding velocity (0.05 m/s), one observes an initial decrease in the friction coefficient down to relatively low values (μ = 0.45) with a subsequent gradual increase up to values in the range 0.61–0.65 upon passing 40 m of the track (curve 1, Figure 9a). Such a behavior of the pair is, possibly, concerned with the fact that at low sliding velocity the formation of the friction track proceeds much slower, i.e., the area of friction pair contact is initially smaller than at higher velocities, and the oxide film formed during friction has a protective function. However, during further friction, the contact area increases, and the role of cold seizure processes, which are more intensive at low sliding velocities, becomes the determining one.

At constant sliding velocity (0.1 m/s) and variable vertical load, one observes similar behavior of friction pairs (Figure 9b): at loads of 10 and 20 N the friction coefficient curve first goes through a maximum (around μ = 0.5) and gradually decreases down to 0.40–0.45. At a load of 5 N, the friction coefficient remains constant within the first 45 m of the track and, thereafter, starts to increase, attaining the values of 0.62–0.63. Such close values of the friction track (curve 1, Figure 9a,b) enable one to assume that on the α → β phase, transition in the titanium sample surface layer in these points of μ increase. The absence of such an effect at other conditions must be explained by more active oxide formation at high values of sliding velocity and vertical load.

Similar data for treated titanium samples are shown in Figure 10.

The linear increase in the friction coefficient is always observed at the friction of titanium samples with a composite Ti-TiC layer in pair with a steel ball at variable sliding velocities and vertical loads. The friction coefficient curves differ only in the slope, depending on the friction conditions. Such a μ increase at the friction of treated samples must be related to the process of steel ball wear. When a steel counterbody rubs on a harder carbide layer, the local increase in temperature in the friction zone is significantly greater than in case of initial samples’ friction because of lower contact areas at the same load (friction track is almost absent (Figure 8c)). This process entails structural changes in the counterbody material, in particular the formation of “friction austenite”, accompanied by an increase in the friction force and wear [38]. As a steel ball wears, the contact area of the friction pair increases (Figure 8d) and, therefore, the friction coefficient increases monotonously (Figure 10a,b).

The presence of numerous scuffs and scratches on the initial samples’ friction tracks (Figure 8a), as well as the titanium transfer on the counterbody surface (Figure 8b), indicates plastic deformation of untreated titanium during friction and the abrasive wear mechanism. After the electric arc processing of the samples, the predominance of adhesive wear is observed, which is expressed in a significant decrease in the width and depth of the friction tracks and in the disappearance of scuffs and scratches (Figure 8c). This significantly increases the counterbody wear (Figure 8d). This behavior is explained by the formation of the Ti-TiC composite structure in the surface layer of the titanium samples, which contributes to an increase in their wear resistance and a decrease in the friction coefficient in a pair with a steel counterbody.

It should be noted that when implementing the arc process, there are difficulties associated with controlling the time of energy supply to the material and the temperature of the material surface volume subjected to plasma exposure. In the future, to control these parameters, as well as to increase the accuracy and repeatability of treatment, it is necessary to use more technological processes, such as laser cladding. A Ti-TiC-based composite material similar in structure can be obtained by fusing titanium carbide powder into the surface of titanium alloys. The study of tribological properties of such material is planned to be carried out in further studies.

## 4. Conclusions

In the process of electroarc discharge treatment of titanium alloys in an aqueous electrolyte using a graphite anode, the titanium-based composite microstructure with TiC grains without oxygen is formed in the alloy surface layer.When moving from the central treatment zone to the periphery, the size and morphology of carbide grains changes from granular (1–10 microns in size) to dendritic (10–150 microns in size). There is free carbon in the amorphous state in the surface layer.The microhardness of the formed composite Ti-TiC-based structure varies from 7.6 to 29.5 GPa; the wear resistance of titanium samples after treatment increases by about 40 times; the friction coefficient of the composite Ti-TiC-layer paired with the steel counterbody gradually increases from 0.08 (min) to 0.3 (max) because of counterbody wear and an increase in the contact area.

## Figures and Tables

**Figure 1 materials-15-08941-f001:**
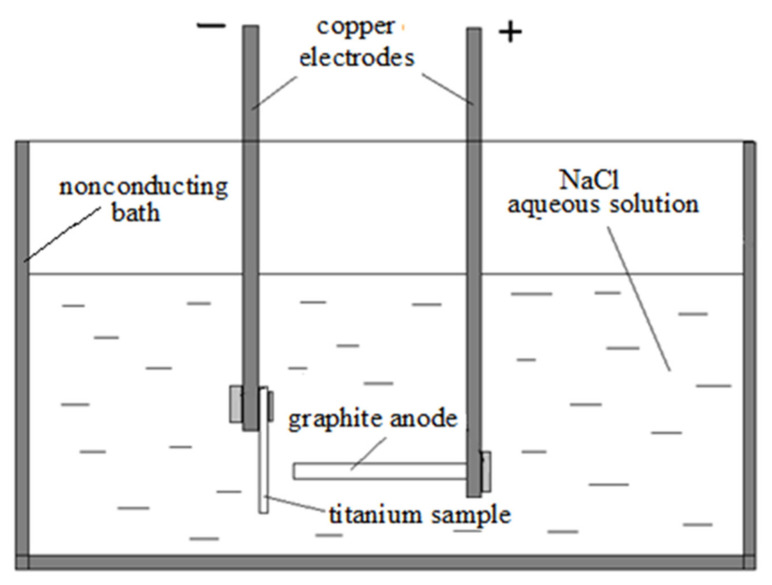
General scheme of samples treatment.

**Figure 2 materials-15-08941-f002:**
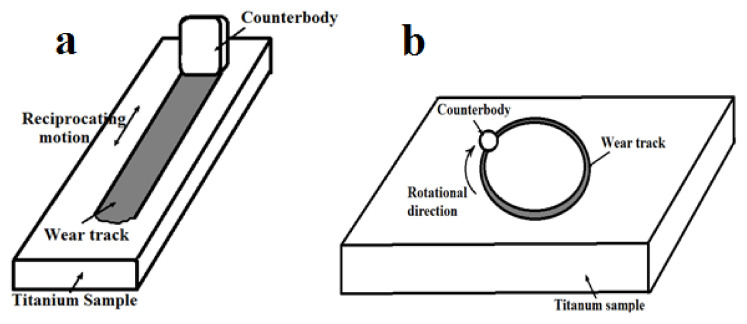
Schematic configuration of samples tests on: (**a**) “pin-on-plate” tester, (**b**) “ball-on-disk” tester.

**Figure 3 materials-15-08941-f003:**
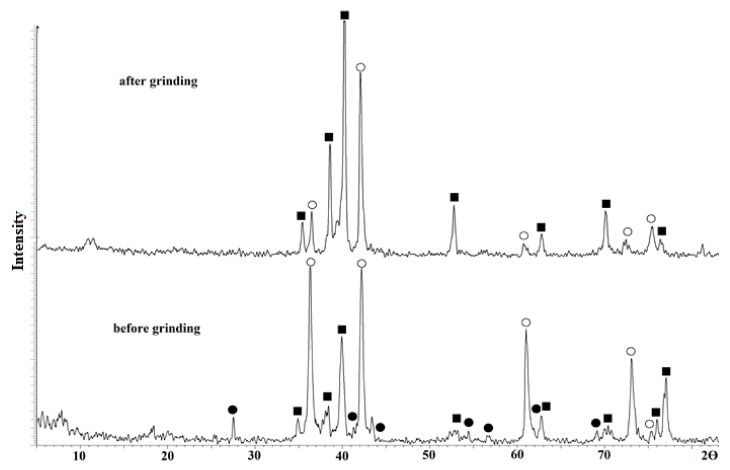
XRD patterns of titanium alloys: ●—TiO_2_ (rutile), ■—Ti, ○—TiC.

**Figure 4 materials-15-08941-f004:**
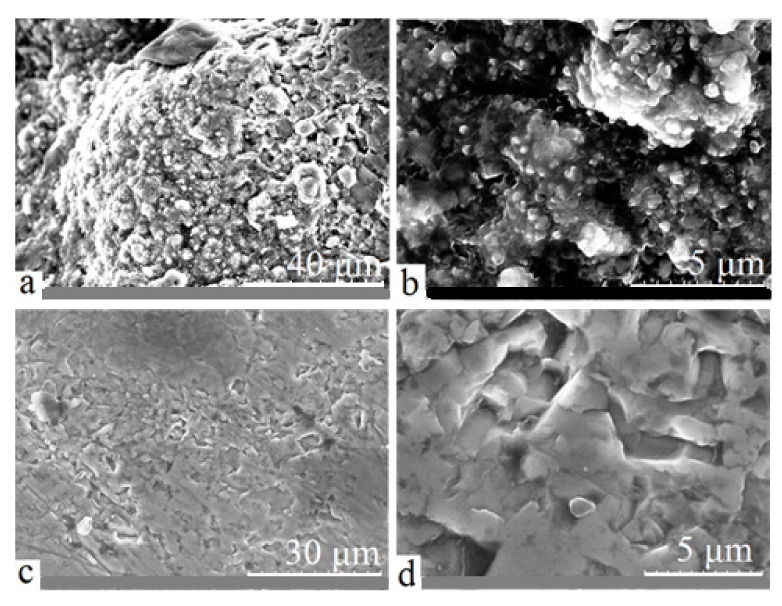
Typical SEM images of treated surface: (**a**,**b**) before grinding, (**c**,**d**) after grinding.

**Figure 5 materials-15-08941-f005:**
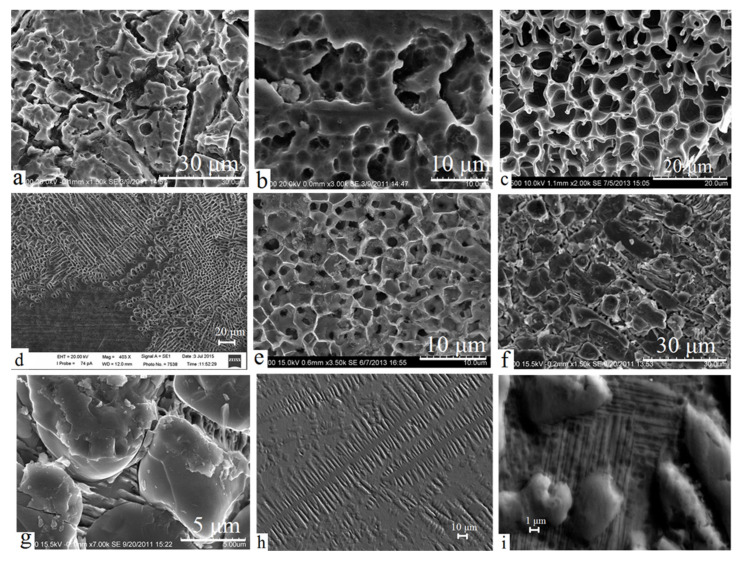
SEM images of the surface after: etching in Kroll’s reagent (**a**,**b**); etching in nitric acid (**c**–**e**) revealing of titanium matrix; etching in hydrochloric acid (**f**,**g**), fine polishing (**h**,**i**) revealing of TiC grains.

**Figure 6 materials-15-08941-f006:**
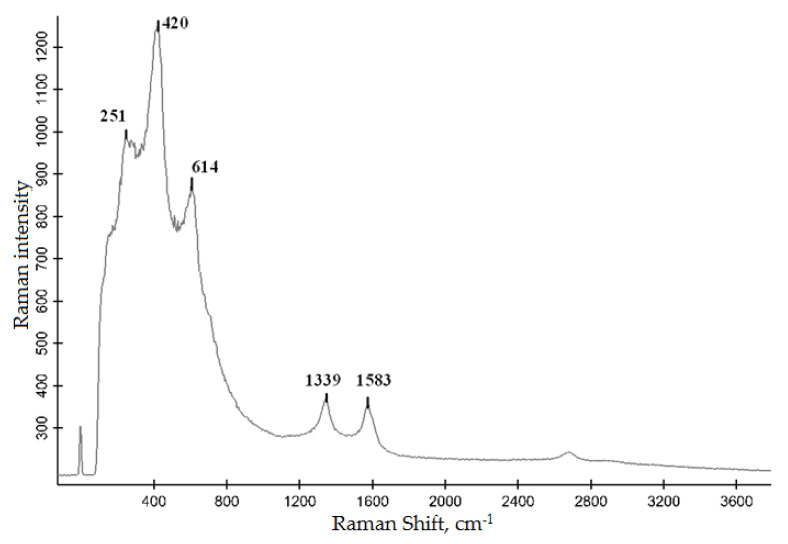
Raman spectra of the surface in local areas of treated samples.

**Figure 7 materials-15-08941-f007:**
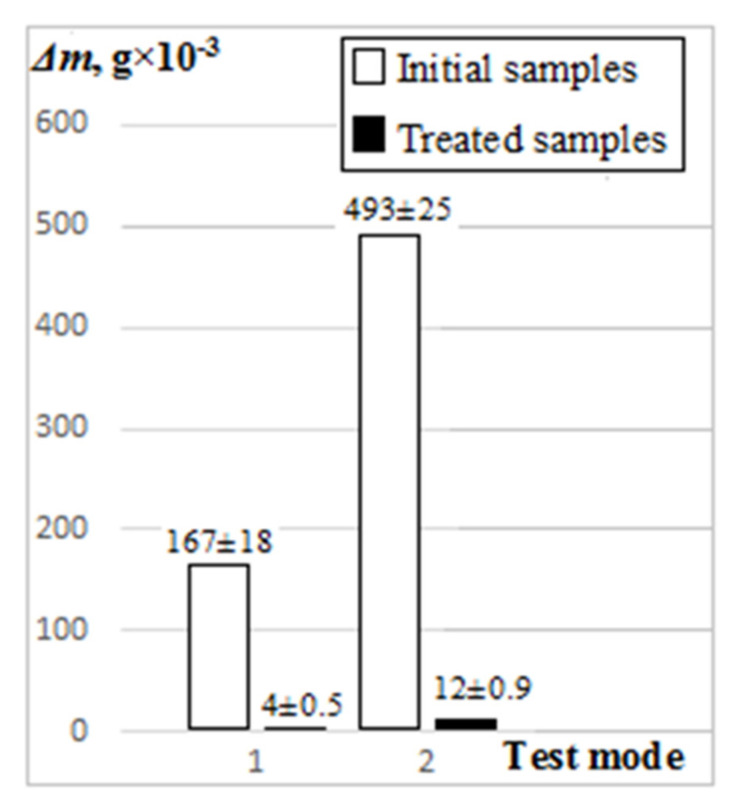
Samples wear in the tests: mode 1: τ = 600 s, P = 30 N; mode 2: τ = 1200 s, P = 50 N.

**Figure 8 materials-15-08941-f008:**
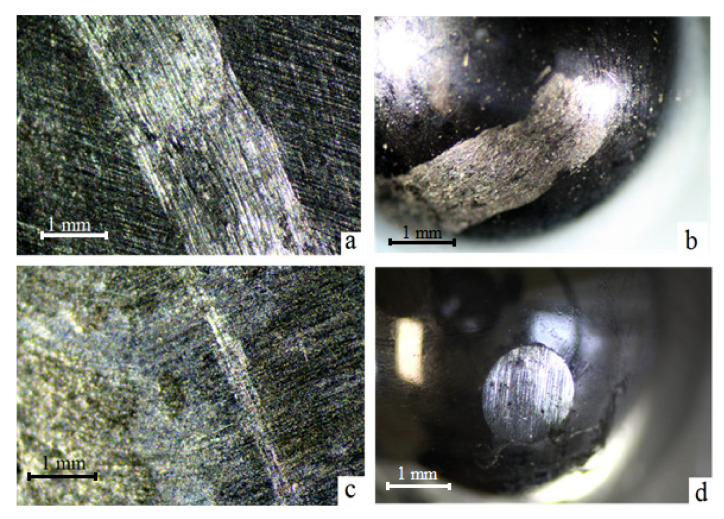
Friction tracks on samples and counter-body after tests: (**a**,**b**) initial samples; (**c**,**d**) treated samples (optical images, ×10).

**Figure 9 materials-15-08941-f009:**
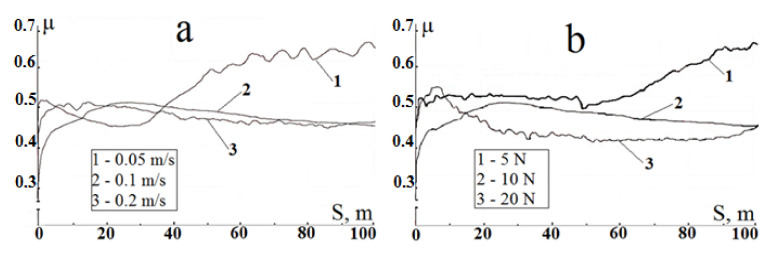
Friction coefficient as a function of sliding distance during the test initial samples: (**a**) at constant load of 10 N and variable sliding velocities 0.05, 0.1, and 0.2 m/s, (**b**) at constant sliding velocity of 0.1 m/s and variable loads 5, 10, and 20 N.

**Figure 10 materials-15-08941-f010:**
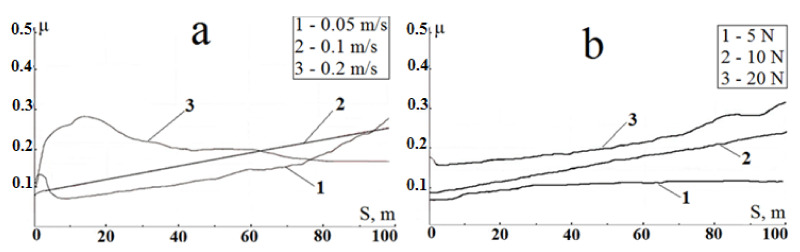
Friction coefficient as a function of sliding distance during the test treated samples: (**a**) at constant load of 10 N and variable sliding velocities 0.05, 0.1, and 0.2 m/s, (**b**) at constant sliding velocity of 0.1 m/s and variable load 5, 10, and 20 N.

**Table 1 materials-15-08941-t001:** Chemical composition at various local sections on the treated surface before and after grinding (at. %).

No.	Before Grinding				After Grinding			
	Ti	C	O	Others	Ti	C	O	Others
1	30.07	32.76	37.16	0.01	63.77	35.54	0.69	-
2	29.87	27.65	42.46	0.02	65.25	34.75	-	-
3	38.03	39.63	22.34	-	64.08	35.92	-	-
4	27.15	4.69	68.15	0.01	66.11	32.59	1.3	-
5	23.69	19.59	56.72	-	61.44	37.34	1.22	-

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
