# Peer review of "Tribological Properties of Ti-TiC Composite Coatings on Titanium Alloys"

_materials, 2022, doi:10.3390/ma15248941_

Round 1

Reviewer 1 Report

The composition, microstructure, and tribological behaviors of the Ti-TiC-based composite coatings formed on titanium alloys were systematically investigated in this manuscript. Comments and suggestions are as follows.

(1) Similar to "pin-on-plate", it is better to use "ball-on-disk" instead of "ball-disk".

(2) Figure 3, there is only one figure, it is not good to use Figure 3a and 3b, curve a and b may be better.

(3) Figure 8, please indicated it is a optical image or SEM image.

(4) Figure 10, there is NO description about the results in figure 10.

(5) It is not good to finish the manuscript after the introduction of friction data, there is too few discussions about the friction and wear mechanism.

Author Response

Thanks for your valuable comments. The responses are in the attached file.

Reviewer 2 Report

The paper presents an interesting approach based on the Tribological Properties of Ti-TiC Composite Coatings on Titanium Alloys. However, the innovation of the current research work should be further highlighted and emphasized. At the same time, the authors should consider the following comments to greatly improve the quality of the paper.

1. In the abstract, add a final statement that highlights the importance of this research and its possible potentials. Also, introduce the problem in the initial lines of the abstract.

2. The introduction needs to be improved by relating to the mechanics of the studied materials and their mechanical characteristics. The references to be included are: 10.1177/07316844211051733, 10.1002/app.46770, 10.1016/j.porgcoat.2022.107015.

3. Kindly add a table that describes the main physical and chemical properties of the raw materials used in this study.

4. Were the preparation methods described by the authors come in accordance with a certain standard or do they follow previous procedures?

5. Why was the microhardness tests used in the testing matrix? and why in the micro scale testing?

6. What is the justification for the specified wear testing parameters in this research?

7. Can you kindly unify the scale bar format for all SEM images?

8. The conclusion needs to be modified to summarize the research outcomes in short statements with clear observations.

Author Response

(The authors gave the same response as above.)

Round 2

Reviewer 1 Report

The revised manuscript has been modified according to the reviewer's comments and suggestions, and it is recommended for publication in Materials.

Reviewer 2 Report

Kindly include the references in introduction: 10.1002/app.46770 and 10.1016/j.porgcoat.2022.107015.